# The FRISK (Fracture Risk)—A New Tool to Indicate the Probability of Fractures

**DOI:** 10.3390/ijerph20021265

**Published:** 2023-01-10

**Authors:** Florian Wichlas, Franziska Melanie Hahn, Serafeim Tsitsilonis, Tobias Lindner, Tim Marnitz, Christian Deininger, Valeska Hofmann

**Affiliations:** 1Department of Orthopedics and Traumatology, Paracelsus Medical University Salzburg, Müllner Hauptstraße 48, 5020 Salzburg, Austria; 2Campus Virchow, Charité University Medicine Berlin, Augustenburger Platz 1, 13353 Berlin, Germany; 3Institute of Tendon and Bone Regeneration, Paracelsus Medical University, 5020 Salzburg, Austria; 4Department of Traumatology and Reconstructive Surgery, BG Trauma Center Tübingen, Eberhard Karls University Tübingen, 72076 Tübingen, Germany

**Keywords:** age, functional questions, fracture risk, visual analog scale, emergency department

## Abstract

Increasing patient inflow into the emergency department makes it necessary to optimize triage management. The scope of this work was to determine simple factors that could detect fractures in patients without the need for specialized personnel. Between 2014 and 2015, 798 patients were admitted to an orthopedic emergency department and prospectively included in the study. The patients received a questionnaire before contacting the doctor. Objective and subjective data were evaluated to determine fracture risk for the upper and lower extremities. The highest risk for fractures in one region was the hip (73.21%; *n* = 56), followed by the wrist (60.32%; *n* = 63) and the femoral shaft (4 of 7, 57.14%; *n* = 7). The regions with the lowest risk were the knee (8.41%; *n* = 107), the ankle (18.29%; *n* = 164), and the forearm shaft (30.00%; *n* = 10). Age was a predictor for fracture: patients older than 59 years had a risk greater than 59.26%, and patients older than 90 years had a risk greater than 83.33%. The functional questions could exclude fractures. Three factors seem to be able to predict fracture risk: the injured region, the patient’s age, and a functional question. They can be used for a probatory heuristic that needs to be proven in a prospective way.

## 1. Introduction

Patient inflow into the emergency department (ED) has increased in recent years [1]. Coping strategies, beyond increasing staff, need to be developed. While extra clinical solutions such as hotlines and general practitioner gatekeepers have been established in some countries, other countries have to optimize their EDs to increase patient outflow [2]. Triage systems such as the Manchester triage, have been developed to identify critically ill or injured patients, not to increase patient flow into the ED [3]. Many decision-making algorithms and heuristics have been implemented since the development of the triage system to address the issue of increased ED inflow [4,5,6].

Orthopedic trauma EDs face similar problems concerning the overwhelming number of patients and the lack of algorithms or tools to increase patient flow sufficiently, except by increasing infrastructure [7]. The main focus of orthopedic traumatology EDs is the detection and treatment of fractures. In most hospitals, a radiological diagnosis must be preceded by an examination by a physician, sometimes even by a specialist. Thus, if a questionnaire or algorithm significantly increases the likelihood of the presence of a fracture, the patient flow could be optimized and the traditional route could be skipped. Similar to modern hospitals where patients are guided by tablets [8] between outpatient departments, in the ED, patients can fill out questionnaires that could sort and direct them efficiently before the first contact with a doctor. There is no doubt that physicians’ roles need to be reduced, starting with simple diagnoses that could be delegated to nurses, telemedicine, or even artificial intelligence [9,10,11].

The goal of this study was to determine simple factors that could detect fractures in patients without the need for specialized personnel.

The hypothesis was that factors exist that could predict a fractured bone or, at least, a high probability of fracture. These factors would result from two different sources from the patient: objective information, such as localization of the fracture and patient age, and subjective information that the patient can articulate, such as pain. Although very different in reliability, both these types of information can be collected.

## 2. Materials and Methods

From 2014 to 2015, we prospectively enrolled 798, 445 men (55.4%) and 356 women (44.6%), with an average age of 42.67 years (SD 20.55), consecutive patients in our orthopedic trauma ED, a level-one trauma university hospital. The inclusion criterion was patients with an extremity mono-trauma; the exclusion criteria were polytraumatized patients and patients without a mechanism leading to a fracture such as stab wounds, pregnancy, and cognitive impairment.

Objective and subjective data were evaluated from questionnaires given to patients in the ED (the questionnaire can be found in the Appendix A). These questionnaires were completed before contact with a doctor and placed in a box. So, the diagnosing doctor never saw the questionnaire and was blinded to it. The level of experience of the diagnosing doctor was 2.23 years (SD 3.42 y; range from 0–6), post-university. The questionnaire was related to the patient and the diagnosis.

For objective data, we evaluated the incidence of upper extremity (UE) and lower extremity (LE) fractures by anatomic region and age.

Subjective data collected included: visual analogue scale (VAS) [12], time of pain onset, and two functional questions for UE and LE:In case of an injury to the UE (questions A1 and A2): Yes or no.○In that moment, were you able to take a bottle of water out of a cupboard with the injured arm? (A1)○In that moment, were you able to untwist a bottle of water with the injured arm? (A2)In case of an injury to the LE (questions B1 and B2): Yes or no.○In that moment, were you able to walk and take four steps without any help, even if there was limping? (B1)○In that moment, were you able to walk and take a few steps on your toes? (B2)

Finally, patients were asked if they thought a bone was broken.

For subjective data, sensitivity, specificity, positive predictive value (PPV), and negative predictive value (NPV) were assessed for the functional questions and patient assessment of a fracture. The functional questions (A1, A2, B1, and B2) were positive for fracture if the answer was “no”. The risk of fracture was assessed for the VAS and compared with the different pain outbreaks using Fisher’s exact test. The radiological examination served as a benchmark. As the questionnaires were adopted during the study, some analyzed subgroups (onset of pain and functional questions A2 and B2) had a lower number of patients. The mechanism of untwisting a bottle (A2) should involve the hand and that of walking on the toes (B2) should involve the foot.

Note that the patients were not supposed to complete questions A and B but just answer them hypothetically.

The study was conducted in accordance with the relevant guidelines and regulations of the Declaration of Helsinki. It was reviewed and approved by the Charité Ethics Committee. Informed consent was obtained from all individual participants included in the study. 

A power analysis for the sample size estimation was performed before study initiation with pain intensity as the primary endpoint with a beta error = 0.80. Differences were considered statistically significant if the null hypothesis could be rejected with >95% confidence (*p* < 0.05). The different parameters (objective and subjective) were analyzed with the Chi-squared test.

## 3. Results

### 3.1. Objective Data

#### 3.1.1. Injured Regions

A total of 798 patients were included, of whom 281 (35.21%) had a fracture. 

More patients with injuries to the LE (*n* = 474; 59.40%) presented in the ED than those with injuries to the UE (*n* = 324; 40.60%), but the fracture risk was higher for the UE (*n* = 150; 46.30%) than for the LE (*n* = 131; 27.64%). This difference was highly significant (*p* < 0.00001, Chi-square test).

The highest risk for fractures in one region was the hip (73.21%; *n* = 56), followed by the wrist (60.32%; *n* = 63) and femoral shaft (4 of 7, 57.14%; *n* = 7); the regions with the lowest risk were the knee (8.41%; *n* = 107), ankle (18.29%; *n* = 164), and forearm shaft (30.00%; *n* = 10) (Table 1).

#### 3.1.2. Age

Half of the patients in the ED were younger than 40 years (*n*= 413; 51.73%) and accounted for one-third of the fractures (*n* = 99; 35.23%). Geriatric patients represented one-fifth of the study population (age > 59 years, *n* = 152; 19.05%) and accounted for one-third of the fractures (*n* = 98; 34.88%). The probability for patients older than 59 years of age to have a fracture was higher than 59.26%; patients older than 90 years had an 83.33% probability of a fracture. The percentage of fractures in patients older than 59 years was significantly higher than that in patients below 59 years (*p* < 0.00001, Chi-square test).

From young to old age, the fracture risk increased almost linearly (Figure 1).

### 3.2. Subjective Data

#### Functional Questions

The sensitivity, specificity, PPV, and NPV were evaluated for anatomic regions (Table 2). The regions where the number of patients was too low were excluded (femoral shaft, *n* = 7; tibial shaft, *n* = 25; humeral shaft, *n* = 8; and forearm shaft, *n* = 10) for this evaluation.

### 3.3. VAS (n = 795)

The VAS values throughout the study population and for the fractured patients are shown in Figure 2. A normal distribution seemed to be apparent, with the highest number of patients having a VAS value of five among all patients in the ED. Most patients with fractures had a VAS score of seven. 

For VAS values higher than six, the risk for fracture was higher than 40%, and for a VAS value of 10, the risk was 57.14%. The number of patients with VAS values < 3 and VAS values > 8 was too low to draw a clear conclusion (*n* = 11–47). 

### 3.4. Pain Onset (n = 363)

The onset of pain showed no significant difference between patients with and without fractures (Table 3).

### 3.5. Patient Fracture Assessments (n = 798)

The patient’s sensitivity was 58.64%, and the specificity was 71.77%. The PPV was 54.92%, and the NPV was 74.74%. 

## 4. Discussion

Increasing patient influx, staff reductions, and demographic trends must lead to further coping measures for Eds. Automation processes will not eliminate professional medical skills, hands-on or complex diagnostics, nor will they lead to a reduction in medical staff tasks. Our suspicion that Eds are overloaded with patients without significant injuries seems to be confirmed, as only 35.21% of patients actually had a fracture. Excluding missed fractures and soft tissue injuries such as ligament tears, a high percentage of patients are still likely to be left with nothing but pain and temporary functional limitations.

We found two objective and one subjective factor that can be used to detect fractures in the ED: the anatomic region of injury, the age of the patient, and the response to functional questions.

### 4.1. Objective Factors

#### Region/Age

The most frequently injured region was the ankle, hand, and foot. The highest fracture risk was at the hip and wrist, and the lowest was at the ankle and knee. Due to the frequent injury at the ankle with low fracture risk, the Ottawa ankle rules already tried to narrow down the radiographic diagnosis [13].

While patients with hip injuries were comparatively the oldest, those with ankle and knee injuries were young. Although most patients coming into the ED are younger than 50 years, the risk of fracture is significantly higher in older patients, especially for hip fractures. Age above 59 years seems to be a very good predictor for fracture in the ED. It is not possible to say whether the factor “age” or “region” is more important. The hip has the highest fracture risk and represents the oldest patient subgroup (age > 59 years); the wrist has the second highest risk, but the age at risk is 30 years younger. Nevertheless, in the subgroup analysis of each region, the percentage of fractures increased with age. Lack of coordination due to mental degeneration and sarcopenia combined with osteoporosis, altered medication, and malnutrition are known factors that lead to falls and higher fracture rates among older adults [14,15,16]. The data appear to confirm the demographics of orthopaedic trauma patients [17]. These findings speak to the need for fall prevention programs, as falls are often the first symptom in older people who are no longer able to manage their daily lives on their own.

### 4.2. Subjective Factors

#### 4.2.1. Questions

Functional questions have shown by their high sensitivity that they are a suitable tool to exclude a fracture. This was true for both UE and LE. However, the sensitivity was lower in comparison for both hand and foot injuries. It is well known that fractures of the hand or foot are most likely to be missed. Two classic examples, in particular, are cited in the literature: the scaphoid bone in the hand and the Lisfranc joint in the foot [18]. These are often not clearly identified in radiography. Due to the mild symptoms associated with these injuries, further diagnosis by computed tomography or magnetic resonance imaging is often not initially initiated. Therefore, we have attempted to integrate these injuries into the functional questions by including the hand with a twisting maneuver (A2) and the foot with walking on the toes (B2). Among the functional questions, questions A2 and B2 had the highest sensitivity. Although the questions were intended primarily to detect hand and foot injuries, sensitivity increased when these were excluded. Thus, a sensitivity of 100% was shown for the wrist as well as at the LE for the hip, ankle, and knee. An NPV of 100% shows that questions related to a corresponding body region can almost exclude a fracture.

#### 4.2.2. VAS

The recorded VAS score showed a normal distribution. The pain perception of the patient with and without fracture showed no clear differences. Our primary assumption that a fracture was more painful could not be confirmed by the data. The reasons for this are manifold. Non-displaced fractures could cause less pain intensity than displaced fractures. Further cofactors could be stress, anxiety, sociocultural background, or even assessment of faster treatment in the emergency department [19,20,21]. However, the frequency of fractures increased with higher scores, but the curve was flat; for VAS scores < 3 and VAS scores > 8, the number of patients was too small to draw a conclusion. In conclusion, VAS does not appear to be a reliable indicator of fracture detection. Thus, VAS should be critically questioned as a triage tool in EDs. Its reliability in elderly patients has already been questioned [22].

The onset of pain was assessed because we assumed that a fracture causes immediate pain because of tears in the tissues, whereas a sprain causes an inflammatory response that leads to a delay in symptoms. However, this assumption could not be confirmed by the data, as again no correlation was found between the onset of pain and a fracture.

The purpose of this work was to determine objective and subjective data indicative of a fracture. The three factors of the region, age, and functional issues can be used for decision-making processes and algorithms as shown in Figure 3. We propose a probatory heuristic that needs to be tested in a prospective procedure. In addition, the questions and pathways need to be further adapted. 

Such a heuristic is consistent with the evolution of modern medicine, as tasks are transferred from physicians to nurses, who are becoming highly specialized in health care, and earlier tasks are transferred from nurses to nonphysician staff [23,24]. Time management is critical for physicians who are responsible for many tasks, such as in the ED, documentation, or the operating room [25].

The formulation of the functional questions comes from the experience of physicians in the emergency department. This was intended to take advantage of physicians’ experience, as an intuitive triage system has been shown to be equivalent to conventional ones [26].

### 4.3. Limitations

It is problematic to generalize these findings for every ED, but the results may demonstrate the principle that in a given setting, factors can lead to diagnoses. The current design of the study only includes bony injuries, so severe ligamentous injuries must be considered separately. In other countries or hospitals, the regions and the age would certainly be different. The functional questions, on the other hand, could still be appropriate for other places. This means that subjective factors are more robust and versatile than objective factors.

## 5. Conclusions

The hip and wrist should raise high suspicion for fractures, while the ankle and knee have a low risk. Hand and foot injuries are easily overlooked.

For medical staff in EDs, older patients should be prioritized, and diagnostics should be achieved quickly because their fracture risk is high. The factor “age” should be respected in any orthopedic trauma triage model. Functional questions can be used specifically to virtually rule out fractures. These data suggest that transferring physician knowledge to an algorithm-based computer program can help optimize the emergency department workflow. 

## Figures and Tables

**Figure 1 ijerph-20-01265-f001:**
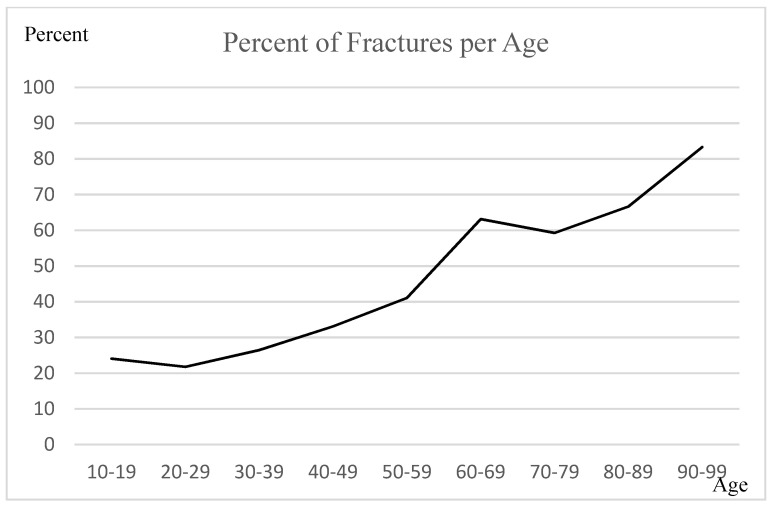
The graph shows the percentage of fractures in groups with increasing age (10-year increments).

**Figure 2 ijerph-20-01265-f002:**
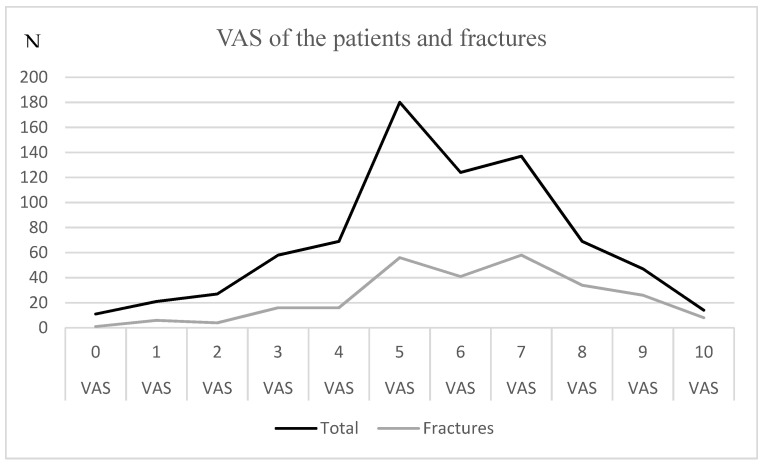
The graph shows the VAS values of all patients (total) and the patients with fractures.

**Figure 3 ijerph-20-01265-f003:**
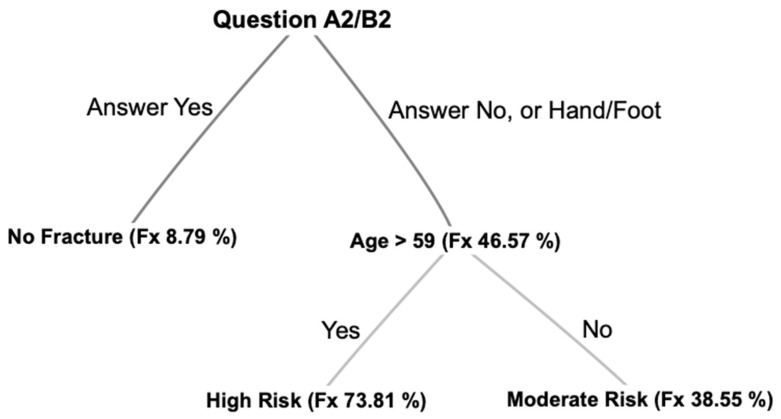
Proposed heuristic.

**Table 1 ijerph-20-01265-t001:** Detailed table of the anatomic regions injured showing the number of patients, the number of fractures and percentages, the mean patient age, and the standard deviation (SD) of the mean age. The highest percentage of fractures and the highest age are highlighted (hip subgroup).

Region	*n*	Fractures	Fractures (%)	Mean Age	SD
Ankle	164	30	18.29	35.18	16.01
Hand	145	66	45.52	36.79	18.46
Foot	116	39	33.62	37.37	14.95
Knee	107	9	8.41	39.50	16.73
Shoulder	69	25	36.23	50.57	18.28
Wrist	63	38	60.32	48.83	22.63
Hip	56	41	73.21	75.39	16.62
Elbow	29	14	48.28	42.83	19.76
Tibia shaft	24	8	33.33	39.67	15.82
Forearm shaft	10	3	30.00	40.70	20.69
Humeral shaft	8	4	50.00	63.50	25.99
Femur shaft	7	4	57.14	69.29	15.18

**Table 2 ijerph-20-01265-t002:** The sensitivity, specificity, PPV, and NPV of questions A1 and A2 for the UE (all, wrist, shoulder, elbow, and hand) and of questions B1 and B2 for the LE (all, hip, ankle, knee, and foot), complete and without hand injuries (A1, A2) and without foot injuries (B1, B2).

	Question	Patients (*n*)	Sensitivity (%)	Specificity (%)	PPV (%)	NPV (%)
Upper Extremity						
All	A1	321	79.22	43.11	56.22	69.23
	A2	144	88.00	31.88	58.41	70.97
Shoulder	A1	68	80.77	40.48	45.65	77.27
	A2	29	85.71	33.33	54.55	71.43
Elbow	A1	29	93.75	53.85	71.43	87.50
	A2	10	100.00	40.00	62.50	100.00
Wrist	A1	58	100.00	40.00	68.75	100.00
	A2	31	100.00	21.43	60.71	100.00
Hand	A1	148	63.89	44.74	52.27	56.67
	A2	67	82.35	36.36	57.14	66.67
All without hand	A1	96	92.68	50.91	58.46	69.23
	A2	77	92.68	27.78	59.38	77
Lower Extremity						
All	B1	473	65.47	67.66	45.73	82.48
	B2	217	95.35	37.40	50.00	92.45
Hip	B1	56	97.56	60.00	86.96	90.00
	B2	41	100.00	57.14	91.89	100.00
Ankle	B1	163	61.29	61.36	27.14	87.10
	B2	66	100.00	29.17	34.62	100.00
Knee	B1	106	75.00	70.40	17.14	97.18
	B2	40	100.00	50.00	18.18	100.00
Foot	B1	115	31.11	74.29	43.75	62.65
	B2	58	85.71	32.43	41.86	80.00
All without foot	B1	362	80.21	66.17	46.11	90.26
	B2	159	98.46	39.36	52.89	97.37

**Table 3 ijerph-20-01265-t003:** Pain onset (Fisher’s exact test).

	*N*	Fractures	No Fractures	*p*-Value
Immediate	287	155	132	0.244
10 min after trauma	37	20	17	0.8627
Within hours after trauma	29	21	8	0.0784
The day after trauma	10	6	4	1.000

## Data Availability

The data presented in this study are available on request from the corresponding author. The data are not publicly available due to privacy.

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
