# Peer review of "The FRISK (Fracture Risk)—A New Tool to Indicate the Probability of Fractures"

_ijerph, 2023, doi:10.3390/ijerph20021265_

Round 1

Reviewer 1 Report

Congratulations to the authors of they work.   Introduction: is comprehensive and well written.  Material and methods: is good and well written, prospective study design increases the educational soundness. Results:is is comprehensive and well written Discussion: is comprehensive and well written   I think your work states what we actually know. The prevalence of common trauma in adults has already been stated by many authors. Idea of facilitating work at ED, especially nowadays is crucial. Any tool or quetionnaire which help optimazing triage management will be benefit.  I think the FRISK questionnaire based on actually two question for upper extremity and two question for lower extremity is not sufficient for triage management but I believe it could be the basis for further questionnaire progression.   I believe the study is worth publishing.   Kind regards.

Author Response

Dear reviewer,

thank you very much for your time and comments. We added some of your notes.

Best reagrds

The authors

Reviewer 2 Report

Due to the increasing world population, patient admissions to emergency services are increasing day by day. This is an important problem for everyone, especially us health professionals.

I would like to thank all the authors for addressing this important issue and for developing solutions.

The paper is beautifully designed and written. The results are very clearly stated and interpreted.

However, issues such as the development and applicability of this study in the future have not been emphasized much. For example, as mentioned in this study, easy-to-apply questionnaires can be used for the development of "Medical Literacy".

Author Response

Dear reviewer,

thank you very much for your time and comments.

Best regards

The authors

Reviewer 3 Report

The aim of the presented paper is not explicit determined. A hypothesis is not stated.

Regarding the methods of the presented paper the authors describe:

1.       prospective inclusion

2.       collection of questionnaires in a box before contact with a doctor

The authors did not describe:

1.       if the diagnosing doctor in the ED was blinded to the questionnaire

2.       the level of experience of the diagnosing doctor

3.       the matching of questionnaire and diagnosis

The authors wanted to present their suggestions for an anamnesis questionnaire to be filled out by ED patients to determine the presence of a fracture including injured region, age, pain, onset of pain and some (hypothetical) questions on the remaining function.

Such a questionnaire could be helpful to split ED-patients in “probably fracture patients” and “probably no-fracture patients” before they see a diagnosing doctor (if they are able to fill out the questionnaire: age, dementia, language barrier, emotional upset etc.).

The number and diagnosis of patients with “soft tissue injuries” is inadequate described.

The presented results are neither surprising nor new:

fractures  were found to be more often in elderly patients, hip and wrist fractures to be more common, higher pain in limitation of function to be suspicious for a fracture.  This knowledge is expected from every nurse or medical student.

The benefit of the paper stays unclear, as the main priority in an ED remains to differentiate between patients who need treatment and those who do not, preferably in the correct priority. According to the specificity in table 2 these parameters are not even given for the group of “probably fracture patients”.

Altogether the paper needs revision.

The methods have to be presented comprehensible.

The possible benefits should be worked out in detail considering an average group of ED patients.

Author Response

(The authors gave the same response as above.)

Round 2

Reviewer 3 Report

thanks for clarifications, paper improved,